# Pin1 Regulates IL-5 Induced Eosinophil Polarization and Migration

**DOI:** 10.3390/cells10020211

**Published:** 2021-01-21

**Authors:** Zhong-Jian Shen, Jie Hu, Melissa A. O’Neal, James S. Malter

**Affiliations:** Department of Pathology, University of Texas Southwestern Medical Center, Dallas, TX 75390, USA; Jie.Hu@UTsouthwestern.edu (J.H.); Melissa.Oneal@UTsouthwestern.edu (M.A.O.)

**Keywords:** eosinophils, asthma, IL-5, allergy, Pin1, shape, migration, polarization

## Abstract

Eosinophils become polarized in response to cytokines such IL-5 or eotaxin prior to directional migration. Polarization is preceded by F-actin assembly, but the mechanisms that regulate these events and how the shape change influences cell migration from the peripheral blood into the lung remain unclear. In this study, we show that the prolyl isomerase, Pin1, is required for IL-5-induced Eos polarization and migration. Co-immunoprecipitation and immunofluorescence analysis revealed that Pin1 directly interacts with members of Rho GTPase family. Mouse eosinophils lacking Pin1 or human cells treated with Pin1 inhibitors showed significantly reduced IL-5-induced GTPase activity and cofilin phosphorylation, resulting in reduced F-actin polymerization, cell polarization, and directional migration to chemokines. Our result suggests that Pin1 regulates cytoskeletal re-organization, eosinophil morphology, and cell migration through the modulation of Rho GTPase activity. Targeting Pin1 along with GTPases could provide a new approach to reduce pulmonary Eos accumulation during asthmatic exacerbations.

## 1. Introduction

Eosinophils (Eos) are the major inflammatory component in allergic asthma, eosinophilic esophagitis and hyper-eosinophilic syndrome [1]. During allergic inflammation, IL-5 induces Eos differentiation, peripheral release, and along with GM-CSF and IL-3 supports cell survival through Jak2-STAT1/5 and Syk-Lyn-Ras-Raf1-ERK1/2 pathways. As a prelude to bone marrow release or pulmonary entry, IL-5 induces Eos shape changes, polarization and transmigration [2,3]. These events require F-actin reorganization and microtubule dynamics, likely facilitating vascular arrest, migration, activation, degranulation and cell survival in inflamed tissues. However, the mechanisms that regulate Eos shape changes prior to chemotaxis-induced cell motility, as well as the ensuing migration, remain incompletely understood.

Rho family GTPases (Rho, Rac and CDC42) play a crucial role in the reorganization of actin cytoskeleton in many cell types [4]. The family includes 23 members divided into several subgroups, including the Rho homologues (RhoA, RhoB and RhoC) and subfamily (Rac1, Rac2 and CDC42), which were found to be distinct in function from other Rho proteins but share significant homology in the amino acid sequence [4]. Rho GTPases act as molecular switches that relay extracellular signals into intracellular events through downstream effectors. In response to stimuli, Rho GTPases undergo a conformational switch from a GDP-bound inactive state to an active, GTP-bound state. Guanine nucleotide exchange factors (GEFs) promote the activation of GTPases by stimulating the exchange of GDP to GTP, while GTPase-activating proteins (GAPs) accelerate the hydrolysis of GTP, returning the GTPases to an inactive form.

Downstream targets of RhoA/B/C include the serine/threonine kinase ROCK, which is mainly involved in the formation of stress fibers and focal adhesions [4]. ROCK is known to phosphorylate the myosin light chain (MLC), leading to actin–myosin contractility. LIMK is another downstream kinase effector of Rho whose phosphorylation of the actin-severing protein cofilin inhibits its function [5]. CDC42 and Rac1/2 along with their scaffold proteins (WASP/SCAR/WAVE) [6] regulate the Arp2/3 complex that mediates actin polymerization at the leading edge of the cells [7,8]. Although considerable progress has been made in understanding the role of GTPases and their effectors in physiologic settings, very little is known about the role of these proteins or their regulation in asthma. Recent RNA Seq and proteomic studies showed that human Eos express high levels of Rho GTPases [9], suggesting a possible role in Eos polarization and migration.

Previously, we showed that germline Pin1 KO or systemic pharmacologic blockade selectively attenuates Eos accumulation in the lungs in animal models of acute airway inflammation [10,11,12]. Pin1 was required for IL-5R-mediated survival signaling and Epstein-Barr virus-induced receptor 2 (EBI2)-induced transmigration of human blood Eos [13,14], suggesting a role for Pin1 in both the entry and enhanced pulmonary survival of Eos during asthma exacerbations. In vitro, Pin1 isomerase activity is rapidly increased by prosurvival cytokines (IL-5 and GM-CSF) as well as in vivo after allergen challenge [13,15]. Pin1 is a highly conserved (from yeast to humans) cis-trans peptidyl-prolyl isomerase (PPIase) that specifically interacts with a subset of intracellular proteins that contain phosphorylated Ser-Pro or Thr-Pro dipeptide motifs (pS/pT-Pro) [16]. The N-terminal Ser or Thr is the target for MAPK, PKC, GSK, and other so-called proline-directed kinases which are commonly activated by cytokine-mediated signaling. Once bound through its N-terminal WW domain, the C-terminal catalytic domain of Pin1 isomerizes the pS/pT-Pro peptide bond to cis/trans conformers, thereby altering the target protein structure, function, and catabolism.

In this study, we explore the potential role for Pin1 in IL-5-induced polarization and migration of mouse and human Eos. We found that Pin1 directly interacted with Rho GTPase, a key molecule mediating cell polarization and shape change. Pin1 blockade reduced F-actin polymerization, likely through changes in the phosphorylation of downstream cofilin. These data suggest the manipulation of this pathway could lead to new therapies for eosinophilic disorders.

## 2. Materials and Methods

### 2.1. Reagents

Anti-Pin1 (G8) (for WB) and anti-cofilin (for Flow) were purchased from Santa Cruz Biotechnology (Dallas, TX, USA). Anti-Pin1 (MAB2294) (for IP) were purchased from R&D (Minneapolis, MN, USA). Anti-β-actin was from Sigma-Aldrich (St. Louis, MO, USA). Anti-Rho A (for IF), anti-GFP (for WB) and anti-phospho-cofilin (Ser3) (for Flow) were from Cell Signaling. Protease Inhibitor Mixture was from Calbiochem (San Diego, CA, USA). Conjugated Myc-DyLight 800 and HA-DyLight 680 were from Thermo Fisher Scientific (Waltham, MA, USA). Human and murine IL-5s were from PeproTech (Rocky Hill, CT, USA). Phalloidin-iFluor 594 Reagent was from Abcam (Cambridge, MA, USA), and the CytoSelect 24-Well Cell Migration Assay was from Cell Biolabs, Inc. (San Diego, CA, USA). Rac inhibitor and Rho inhibitor/Rhosin were from Calbiochem (San Diego, CA, USA), and RhoA G-LISA Activation Assay Kit was from Cytoskeleton Inc. (Denver, CO, USA).

### 2.2. Mice

Pin1^fl/fl^ mice were created on a pure C57Bl6 background. Exon2 is flanked by two loxP sites. The floxed mice were bred with EoCre mice (C57Bl6) (from Dr. J. J. Lee, Mayo Clinic) to delete Pin1 in the eosinophils. Animal care was carried out in accordance with the Guide for the Care and Use of Laboratory Animals of the NIH. Our animal protocol was approved by the Committee on the Ethics of Animal Experiments of the UTSW-Medical Center (Permit no. 2011-0139).

### 2.3. Human Eos Preparation

Peripheral blood was obtained from healthy or mildly atopic donors. Eos from the peripheral blood were purified as described previously [17], and populations of >96% Eos were used in the studies. Cells were cultured at 1 × 10^6^ cells/mL in RPMI 1640 medium plus 10% FBS. Written informed consent from all participants was obtained according to an approved protocol of the University of Wisconsin Hospital Institutional Review Board. The viability of cells after purification was >96%.

### 2.4. Immunostaining, Immunoblots and Intracellular Flow Cytometry

For the analysis of Pin1 and RhoA localization, cytospun Eos were incubated with primary Abs followed by FITC-conjugated secondary Abs. Images were collected with at least six fields per slide. For immunoprecipitation (IP) and immunoblots of recombinant proteins, two myc-tagged proteins were mixed in Nonidet P-40 buffer and incubated with normal IgG or anti-Myc Ab followed by pull-down with protein-G Sepharose beads (Sigma) (St. Louis, MO, USA). After SDS-PAGE, proteins were transferred onto PVDF membranes and probed with primary and secondary Abs. Protein bands were detected and quantified using the LI-COR Odyssey Imaging System (BioAgilytix, Durham, NC, USA). For intracellular flow cytometry, cells were fixed (paraformaldehyde) and permeabilized (methanol) before co-staining with anti-total and anti-phospho-cofilin followed by secondary antibodies. The specificity of each antibody was confirmed via staining with isotype controls.

### 2.5. Recombinant TAT Proteins

Pin1 inhibitory peptides (TAT-WW and its mutant TAT-W34A) were synthesized and purified at Genemed Synthesis Inc. (San Antonio, TX, USA). The full-length Pin1 and GFP cDNAs were cloned in frame into a pHis-TAT vector for His6-TAT-HA-proteins. The pHis-TAT vector BamHI site between the His tag and the TAT tag was mutated to GGTTCC using PCR mutagenesis. The vector was digested with BamHI and EcoRI to remove the HA tag. A PCR reaction generated BamHI-Myc-RhoA-EcoRI from pRK5-Myc-RhoA (Addgene) (Watertown, NY, USA) and the product was cloned into the modified backbone using the In-Fusion™ HD EcoDry Cloning kit (Takara Bio Inc.) (Kusatsu, Japan) to yield His6-TAT-Myc-RhoA. Proteins were expressed in Escherichia coli and purified on a Ni2^+^ chelate column. The TAT-peptides were more than 90% pure on Coomassie blue staining of SDS-PAGE.

### 2.6. Pull Down Assay

Anti-Pin1 (R&D) was crosslinked to Protein G Sepharose Fast Flow Beads (Sigma) as before. Samples of equal amounts of protein (4 μg) were incubated at 37 °C for 2 h. Anti-Pin1 XL beads were added to the samples for 1 h. Beads were washed, moved to new tubes, and boiled in 1X SDS Loading Buffer.

### 2.7. Cell Polarization and Migration Assay

The width (minor axis) and length (major axis) of the acorn-like cell shape (as a result of polarization) [2] were measured after the cytospin and photographing. At least 50 cells were measured, and the polarization ratio was normalized by dividing the length over the width of the cells. For the cell migration assay, a CytoSelect 24-Well Cell Migration Assay kit was employed. Cells were placed at the top of the insert in a medium with 10% FBS, while chemoattractants were added to the lower chamber. Inhibitors were directly added to the cell suspension before induced migration with chemoattractants. Cells migrated through the pores (5 μm) to the lower chamber were lysed and stained with DNA dye (CyQuant GR) (Waltham, MA, USA) prior to absorbance measurements at 480 nm/520 nm.

### 2.8. Eos Differentiation from Wild Type and Pin1 KO Bone Marrow

Eos were differentiated in vitro from bone marrow (BM) cells as described [18]. BM were collected from long bones by flushing with culture medium. After lysing the RBCs, the remaining cells were cultured at 1 × 10^6^/mL in RPMI 1640 with 20% FBS, 2 mM glutamine, 25 mM HEPES, 10 mg/mL streptomycin, 100 IU/mL penicillin, 13 nonessential amino acids, 1 mM sodium pyruvate, and 50 mM 2-ME, supplemented with 100 ng/mL rmSCF (PeproTech) (Rocky Hill, CT, USA) and 100 ng/mL rmFLT3L (PeproTech) until day 4. On day 4, the media was exchanged with fresh media containing 10 ng/mL IL-5 only. Every other day from day 6, 50% of the media was replaced with fresh media containing 10 ng/mL IL-5. Right before each experiment, dead cells were removed with the Dead Cell Removal Kit (Miltenyi Biotec) (Bergisch Gladbach, Germany), and preparations with >96% viable cells were used for all studies.

### 2.9. Pin1 Activity Assay

Activity was measured as described previously [19] with slight modifications.

### 2.10. Statistics

All *p*-values were calculated via one-way ANOVA using GraphPad Prism 9.0.0. For data comparison between multiple groups, we conducted non-parametric one-way analysis followed by a post-hoc Tukey’s test. Data are represented as mean ± SD. A *p*-value <0.05 was considered statistically significant.

## 3. Results

### 3.1. Pin1 Is Required for IL-5 Induced Eos Polarization

Eosinophils undergo polarization and shape changes followed by transmigration from the blood to tissue compartments in response to chemotactic stimuli including platelet-activating factor, IL-5 and eotaxin-1 [20]. During polarization, CD44 and PSGL-1 redistribute to one pole of the cells along with nucleus, constituting a nucleopod [2]. Morphologically, the cells change from a round to an acorn-like (elongated) shape, which often correlates with an increased forward scatter (FSC) in flow cytometric analysis. Once in tissues, granule proteins are secreted, leading to damage and further inflammation. However, the mechanisms regulating these events in the asthmatic lung remain ill-defined. As Pin1 is required for IL-5R mediated survival signaling and Eos differentiation [11,13], we asked whether Pin1 also plays a role in cytokine-induced polarization. Purified human blood Eos were starved (3 h) followed by treatment with recombinant human IL-5 at different doses (0–50 ng) prior to fixation and morphologic analysis. Pin1 inhibitory peptide (TAT-WW) (100 nM) and its mutant, inactive control (TAT-W34A), were added prior to IL-5 and rapidly entered the Eos cytosol. In published experiments [11,13], this concentration of TAT-WW inhibits Pin1 by >90% while TAT-W34A has no effect. As shown (Figure 1A), in the absence of IL-5, cells remained relatively round irrespective of the Pin1 blockade by TAT-WW or the delivery of the inactive control TAT-W34A peptide. As Pin1 is inactive in resting Eos [12,13], these results suggest TAT-mediated transduction had no observable effect on cell morphology. IL-5 (5 ng/mL) treatment induced polarization in control cells, which was significantly suppressed by the Pin1 inhibitor. ImageJ quantification showed that the ratio of length to width (elongation) of IL-5-treated cells was about 1.30 (mean) in TAT-W34A-treated cells; however, the ratio was reduced to 1.13 after TAT-WW treatment (Figure 1B). The minimum effective dose of IL-5 was 5 ng/mL, which was not significantly enhanced by higher concentrations (10–100 ng/mL). Therefore, IL-5-induced shape change is Pin1-dependent.

Next, we asked whether the data after the acute Pin1 blockade would be seen in Eos derived from conditional Pin1 KO mice. We bred EoCre mice with Pin1^fl/fl^ animals, both in pure C57/Bl6 backgrounds. Pin1 is therefore selectively ablated in differentiating Eos, which was indeed observed based on immunoblotting and flow analysis [11]. We differentiated WT and Pin1 null Eos from bone marrow (BM) in the presence of stem cell factors and IL-5 [18]. Eos developed normally in both genotypes based on the appearance of cell surface markers, viability, yield, and morphology [11]. After the removal of dead cells (10–20% of the total population) from day 15 BM cultures (>98% Eos), the remaining live cells were starved for 3 h at 37 °C before treatment with recombinant mouse IL-5 (50 ng/mL). We obtained similar data (Figure 1C) with mouse KO Eos as human Eos treated with TAT-WW, although the amplitude of the response in WT mouse cells was less than in human Eos. This is likely attributable to the presence of a high dose of IL-5 (10 ng/mL) during the mouse Eos differentiation from day 4 forward.

### 3.2. IL-5 Induction of F-Actin Polymerization Is Pin1 Dependent

Cell polarization is mediated by F-actin-microtubule assembly, which along with associated proteins generates a wave-like movement [21]. In Eos, the F-actin network may also be responsible for granule integrity, subcellular localization, and organization as activated Eos migrate into and within inflamed tissues. Therefore, we asked if Pin1 was involved with IL-5 driven, F-actin polymerization in human peripheral blood Eos. Cells were starved and then incubated with Pin1 inhibitor (TAT-WW) or control peptides (TAT-W34A) for 3 h before treatment with IL-5 for 10 min. After fixation, cells were stained with phalloidin for imaging and quantitation of F-actin via flow cytometry. Unlike fibroblasts and epithelial cells, where the actin cytoskeleton is organized in long and numerous stress fibers regardless of the substrates used for attachment (poly-lysine, fibronectin or collagen), Eos lack these typical structures irrespective of cell activation. This is likely due to embedded granules in the F-actin network, a unique mechanism to control granule integrity and organization. While the phalloidin-positive stress fibers were imaged as localized throughout cytoplasm, they were not significantly affected by cell activation or/and Pin1 blockade (Figure 2A). However, we observed that IL-5R stimulation for 10 min significantly increased the mean fluorescence intensity (MFI) of F-actin in TAT-W34A treated human and mouse Eos (Figure 2B,C), suggesting that actin polymerization is induced and remodeled by IL-5 signaling. Notably, this induction was suppressed after Pin1 blockade with TAT-WW (Figure 2B) or in KO mouse Eos (Figure 2C), suggesting that IL-5R stimulation induces actin polymerization and remodeling in a Pin1-dependent manner.

### 3.3. Pin1 Is Required for IL5-Induced Cell Migration

Actin remodeling plays a crucial role in a variety of cellular processes required for normal immune function, including intercellular interactions, endocytosis, cytolysis, cytokinesis, signal transduction, and maintenance of cell morphology [22]. IL-5 acts as a strong chemokine and survival factor for Eos and also induces Pin1 isomerase activity [11,13]. Conversely, Pin1 blockade suppresses IL-5 induced Eos survival through the activation of pro-apoptotic Bax [15]. Therefore, we asked whether Pin1 blockade can also suppress IL-5-induced Eos migration. Peripheral blood Eos were pre-incubated with TAT-W34A or TAT-WW (100 nM as above) for 1 h before placement in the upper trans-well chamber, while IL-5 was added to the lower chamber. After incubation for 18 h, migrated cells were lysed and quantitated. In the presence of 10% fetal bovine serum, IL-5 (50 ng/mL) alone or with control peptide TAT-W34A induced significant migration into the lower chamber which was, however, completely blocked in Eos treated with TAT-WW (Figure 3A). Of note, we observed that mature, BM-derived Pin1 KO mouse Eos also failed to migrate, suggesting a crucial role for Pin1 in mediating IL-5 induced Eos polarization and migration (Figure 3B). Cell viability over the experimental time course was equivalent between conditions (Figure 3C), eliminating differential apoptosis as a cause for the observed differences. These results are largely consistent with post-allergen-challenge in vivo events in which pulmonary Eos accumulation was reduced in Pin1 KO mice, although a lower IL-5 level in KO could also conceivably contribute.

### 3.4. The Effect of Rho Inhibitors on Eos Polarization and Migration

Rac2, a hematopoiesis-specific Rho GTPase, was shown to regulate F-actin formation, shape change, O2^−^ release, and degranulation in Eos in response to eotaxin-2 or platelet-activating factor (PAF) [23]. Eos express a high level of multiple Rho GTPases, including RhoA, Rac2, and CDC42, as well as the F-actin effector, cofilin [9,24]. Thus, we hypothesized that GTPases regulate IL-5-mediated actin-remodeling and downstream polarization and migration. To test, we pre-incubated human blood Eos with specific GTPase inhibitors prior to IL-5 treatment and followed subsequent polarization and migration compared to untreated controls. Interestingly, both RhoA and Rac inhibitors inhibited polarization (Figure 4A) and migration (Figure 4B) at similar doses. These data confirm that Rho GTPases likely mediate IL-5 signaling to control cell shape, polarity and migration.

### 3.5. Pin1 Interacts with Rho GTPase and Regulates Downstream Signaling

The lack of IL-5-induced polarization after Pin1 blockade/deletion or inhibition of Rho GTPases suggested a common mechanism of action. Therefore, we asked if Pin1 and the GTPases interacted. Sequence analysis revealed putative pS/pT-Pro sites that could be recognized by Pin1 in all Rho family members. We co-incubated recombinant proteins Pin1 with GFP or RhoA. Pull-down followed by immunoblots revealed a direct interaction between Pin1 and RhoA (Figure 5A). Immunofluorescence staining of Eos with anti-Pin1 and anti-RhoA showed colocalization of the two molecules in untreated cells, which was increased after IL-5 treatment (Figure 5B). Interestingly, two canonical Pin1 binding sites in RhoA (S88-P89 and T100-P101) are phosphorylated after cell activation and contribute to cytoskeletal reorganization [25]. While we have not formally demonstrated it, we hypothesize Pin1 binds to and likely isomerizes RhoA at one/both of these sites, presumably regulating GTPase activity. It remains to be investigated whether Pin1 also interacts with other GTPases (e.g., Rac and CDC42). Based on the above data, we propose Pin1 loss/inhibition or RhoA blockade prevents F-actin assembly and Eos polarization and migration.

As Rho-family GTPases ultimately regulate F-actin through cofilin [26,27], we rationalized that IL-5 and Pin1 might also utilize cofilin to affect F-actin content. WT and Pin1 KO Eos were stimulated with IL-5 before staining with anti-pSer3-cofilin and anti-total-cofilin. Brief treatment (10 min) of WT cells with IL-5 significantly increased cofilin phosphorylation, which was completely blocked in the absence of Pin1 (Figure 5C). The levels of total cofilin were unchanged in either genotype despite IL-5 treatment. These data suggest that Pin1 regulates Eos motility, likely via GTPase-cofilin pathways in response to IL-5 stimulation, presumably via direct interactions with RhoA.

### 3.6. Pin1 Regulates Rho GTPase Activity

The Pin1-RhoA interaction suggested possible uni- or bidirectional regulation. To clarify this, we individually inhibited Pin1 or RhoA and then measured the other’s activity after Eos stimulation with IL-5. To block Pin1, human Eos were pre-incubated with control TAT-peptide (W34A) or TAT-WW for 1 h before brief treatment (10 min) with IL-5. As shown (Figure 6A), IL-5 significantly increased RhoA activity in cells pre-incubated with control peptide which was blocked by TAT-WW. Conversely, IL-5 rapidly activated Pin1 isomerase activity (Figure 6B) [13], which was unaffected by either a Rac or Rho inhibitor (Figure 6B). These data suggest Pin1 regulates RhoA rather than the reverse and predicts that a Pin1 dependent conformational change of Rho GTPases is required for downstream signaling events and cytoskeleton reorganization (Figure 6C).

## 4. Discussion

Eosinophils undergo shape changes, polarization, and ultimately migration in response to chemotactic gradients created with platelet-activating factor, IL-5 and eotaxin-1. These alterations are a prelude to transmigration from the blood to tissue compartments and granule protein secretion [20]. Despite their fundamental importance, the molecular mechanisms underlying morphologic reorganization and cell migration have remained unclear. In this study, using inhibitors and genetic knockdown approaches, we demonstrate for the first time that Pin1 is required for cell polarization and migration induced by IL-5 in both human and mouse Eos. Through direct interactions, Pin1 likely regulated the activity Rho GTPases in response to IL-5 with the degree of phosphorylation of cofilin, the ultimate downstream event that altered F-actin polymerization and presumably cytoskeleton dynamics.

Rapid shape change induced by IL-5 often precedes Eos migration in which MAPK activation is an important upstream mediator [2,3]. We found that IL-5 also activates downstream Rho GTPases (Figure 6A), while others have shown that CCL11/eotaxin-1 can activate RhoA, Rac, and ERK MAPK, leading to actin polymerization and granule secretion in Eos [23,28]. Whether GTPases and MAPKs cross-regulate each other in Eos or other cell types remains to be determined. In other cells, GTPases have been implicated in the activation of the MAPK pathway. For example, in fibroblasts or hematopoietic stem cells, alteration of CDC42 activity selectively abrogated JNK MAPK activity without impairing ERK and p38 MAPK signaling in response to serum or growth factors [29,30]. Whether similar events exist in Eos is unknown.

Eos express multiple GTPases at high levels [9,24]. However, the contribution of individual GTPases to Eos effector functions is largely unknown. Some of the knowledge gaps reflect either unavailable or nonspecific GTPase inhibitors, and few investigators have developed Eos-specific conditional KO mice to selective query the roles of GTPase in these cells. One such study showed Eos from Rac2 KO mice had diminished degranulation and shape changes in response to eotaxin-2 or platelet-activating factor compared to wild-type cells [23]. Mice lacking Rab27a, which is highly expressed and activated in asthma [31], had reduced EPX release in BAL fluid, again supporting roles of GTPases in Eos exocytosis and degranulation. The function of GTPases is likely to be complex, and each GTPase may play distinct but interconnected roles in actin reorganization. Cell polarization and migration were relatively sensitive to the Rho inhibitor compared to those for Rac (Figure 4A). In fibroblasts, Rac was required for the formation of membrane ruffles, whereas Rho was required for the formation of stress fibers [32,33]. Activation of CDC42 in Swiss 3T3 cells led to sequential activation of Rac and Rho, suggesting specific and highly coordinated control of cell motility by multiple GTPases [34] which are highly cell-type specific.

The role of Pin1 in the control of GTPase activity and actin polymerization has not been studied elsewhere. However, Pin1 has been implicated in EBI2-induced Eos migration [14] and TGF-beta-induced migration of tumor cells [35]. Recombinant Pin1 directly interacted with RhoA GTPase and blockade of Pin1 in Eos suppressed intracellular RhoA activity normally induced by IL-5. Notably, the RhoA blockade did not affect Pin1 activation or activity, indicating unidirectional, post-translational control of RhoA GTPase by Pin1. In addition, Eos lacking Pin1 showed significantly reduced expression of GTPases at both mRNA and protein levels during cell differentiation from BM cultures (unpublished data), suggesting Pin1 also supports the transcription of GTPases. We have struggled to pull down endogenous Pin1 with RhoA in cell lysates despite the above data, suggesting a weak or transient interaction in live cells or artifactual disruption during cell lysis. RhoA has two conserved Pin1 sites (Ser88-Pro89 and Thr100-Pro101) that are phosphorylated by ERK MAPK in response to EGF stimulation [25]. Phosphorylated RhoA is more active and is associated with increased actin fiber formation in COS-7 cells [25]. As Pin1 isomerase activity is increased by IL-5, we hypothesize that Pin1 transiently binds to and isomerizes RhoA, leading to GTPase activation, cofilin inactivation and F-actin assembly and cell polarization.

These results also implicate known downstream effectors of RhoA signaling but also those upstream of cofilin as likely involved. These include LIM kinase, which phosphorylates cofilin on Ser3 in response to contractile stimuli, thus preventing F-actin depolymerization [4,5,26]. Cofilin acts by directly binding to actin filaments and thus is essential for cellular processes that require rapid actin filament turnover such as cell motility, cytokinesis and endocytosis. Eos express cofilin and its effectors such as Arp2/3 [9,24]. We found that IL-5 itself significantly increases cofilin activity which was, however, completely blocked in the absence of Pin1. These data suggest that Pin1 regulates Eos motility, likely via GTPases-LIM kinase-cofilin pathways in response to IL-5 stimulation.

These results are not entirely unexpected as Pin1 has been shown to regulate several cytokine-driven signaling events in Eos. These include the expression and release of pro-/anti-inflammatory cytokines (e.g., GM-CSF, TGF-β, and IFN-α/β) and granule proteins (MBP and EDN) in response to allergic mediators and respiratory viral infection [10,11,12,13]. Pin1 was required for Eos differentiation and maturation in the bone marrow, as well as peripheral blood Eos survival and transmigration in allergic asthma [11,13,14]. Mechanistically, Pin1 typically interacted with important effectors such as RNA binding proteins (AUF1) kinases (PKC, IRAK4, ERK1/2), and apoptotic pathway proteins (Bax, Caspase 8). The regulation of Rho GTPases and their signaling illustrates and extends the vital and pleiotropic role for Pin1 in Eos function and, by extension, the pathophysiology of asthma and possibly other eosinophilic pathologies.

In conclusion, our findings provide new evidence supporting the role of Pin1 in Eos polarization and migration both in human and mouse cells. These findings also expand the potential role of RhoA GTPases in Eos biology and potentially asthma pathophysiology. Further studies in this area should continue to yield new and important discoveries relevant to Eos function.

## Figures and Tables

**Figure 1 cells-10-00211-f001:**
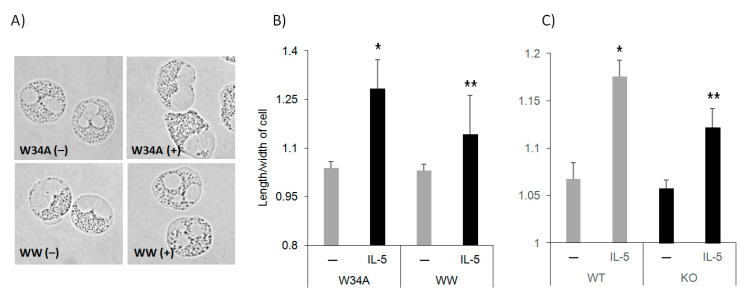
Pin1 is required for IL-5-induced Eos polarization. (**A**) Human peripheral blood Eos (>98% pure, >95% viable) were starved for 3 h prior to incubation with Pin1 inhibitor (TAT-WW, 100 nM) or control peptides (TAT-W34A) for 1 h before treatment with hIL-5 (50 ng/mL) for 10 min. (**B**) The length and width of cells from (**A**) were measured by ImageJ. (**C**) BM-derived mouse Eos were starved (3 h) before treatment with mIL-5 (50 ng/mL) for 10 min. Data from three independent experiments. * *p* < 0.05 between untreated (–) and IL-5 in W34A-treated cells, ** *p* < 0.05 between IL-5 treatments.

**Figure 2 cells-10-00211-f002:**
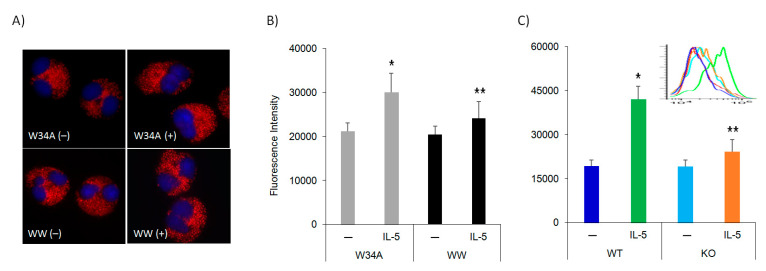
IL-5 induction of F-actin polymerization is Pin1 dependent. (**A**) Human peripheral blood Eos were starved and incubated with Pin1 inhibitor (TAT-WW, 100 nM) or control peptides (TAT-W34A) for 1 h before treatment with hIL-5 (50 ng/mL) for 10 min. Cytospun cells were stained with phalloidin before imaging. (**B**) Fluorescent intensity of cells from (**A**) were measured by flow cytometry. (**C**) BM-derived mouse Eos were starved (3 h) before treatment with mIL-5 (50 ng/mL) for 10 min. Cells were stained with phalloidin and the intensity was measured using flow cytometry. Data are from three independent experiments. * *p* < 0.05 between untreated (–) and IL-5 in W34A-treated cells, ** *p* < 0.05 between IL-5 treatments.

**Figure 3 cells-10-00211-f003:**
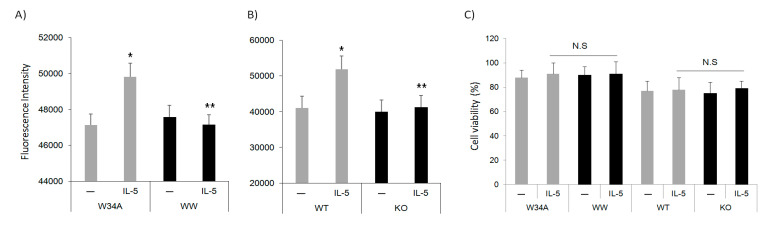
Pin1 is required for IL-5-induced cell migration. (**A**) Human peripheral blood Eos were starved and incubated with TAT-WW (100 nM) or control peptides (TAT-W34A) for 1 h before transmigration for 18 h towards hIL-5 (50 ng/mL). (**B**) BM-derived mouse Eos were starved (3 h) before migration for 18 h toward mIL-5 (50 ng/mL) and mEotaxin (20 nM) in the conditional medium. Migrated cells were quantitated by staining with DNA dye. (**C**) Cells were treated as in (**A**) and (**B**), and viability was determined using the trypan blue exclusion method after 18 h incubation. N.S: no significant. Data are from three independent experiments. * *p* < 0.05 between untreated (–) and IL-5 in W34A-treated cells, ** *p* < 0.05 between IL-5 treatments.

**Figure 4 cells-10-00211-f004:**
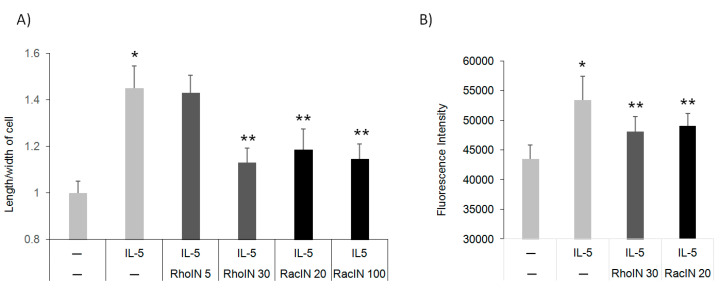
Rho inhibitors suppress IL-5-induced Eos polarization and migration. (**A**) Human peripheral blood Eos were starved and incubated with RhoA (5 and 30 μM) or Rac2 inhibitor (20 and 100 μM) for 2 h before treatment with IL-5 (5 ng/mL) and shape measurement. (**B**) Cells were treated with inhibitors as in (**A**) before the induction of cell migration toward hIL-5 (50 ng/mL). Data are from three independent experiments. * *p* < 0.05 between untreated (–) and IL-5, ** *p* < 0.05 between IL-5 and IL-5+GTPase inhibitors.

**Figure 5 cells-10-00211-f005:**
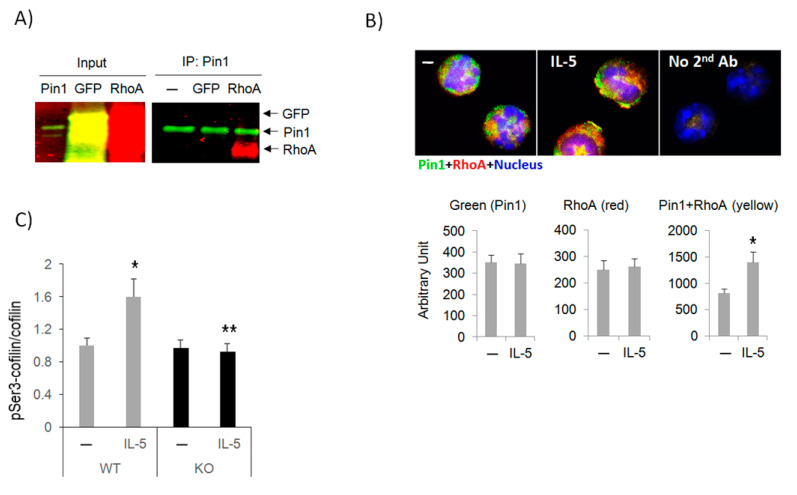
Pin1 interacts with Rho family GTPases and regulates downstream cofilin activity. (**A**) Recombinant proteins were mixed in each condition prior to immunoprecipitation with anti-Myc followed by immunoblot with anti-Pin1 and anti-GFP. (**B**) Human blood Eos were starved for 3 h before treatment with hIL-5 for 10 min. Cytospun cells were co-stained with anti-Pin1 (green), anti-RhoA (red), and DAPI (blue) before examination by fluorescence microscopy and quantification of each color. (**C**) BM-derived mouse Eos were starved for 3 h before treatment with mIL-5 (50 ng/mL). Cells were stained with anti-pSer3-cofilin and anti-cofilin for flow cytometry. Data are from three independent experiments. * *p* < 0.05 between untreated (–) and IL-5 in WT, ** *p* < 0.05 between IL-5 treatments.

**Figure 6 cells-10-00211-f006:**
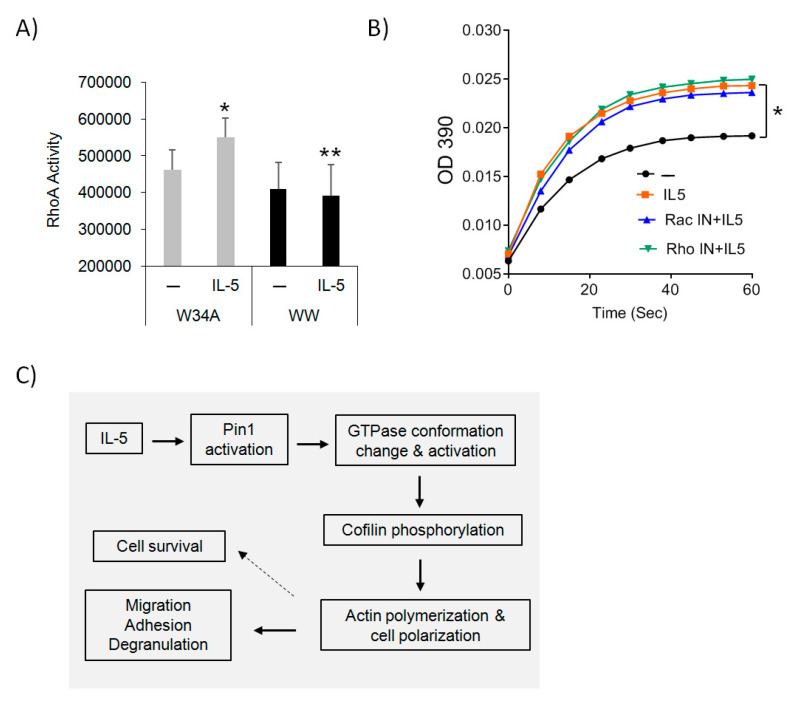
Pin1 regulates GTPase activity. (**A**) Purified human peripheral blood Eos were starved and incubated with TAT-WW (100 nM) or control peptides (TAT-W34A) for 1 h before treatment with hIL-5 (5 ng/mL) for 10 min. The activity of RhoA was measured using commercial G-LISA kit (Cell Biolabs Inc.). Data are from three independent experiments. * *p* < 0.05 between untreated (–) and IL-5 in W34A-treated cells, ** *p* < 0.05 between IL-5 groups. (**B**) Human blood Eos were starved and incubated with RhoA (30 μM) or Rac inhibitor (100 μM) for 2 h before treatment with hIL-5 (5 ng/mL) for Pin1 isomerase activity assay. * *p* < 0.05 between untreated (dark circle) and IL-5 treatment (red square). There were no significant differences between IL-5 and RacIN+IL-5 or RhoIN+IL-5. (**C**) The proposed model for Pin1 regulation of Eos polarization and migration. Pin1 is rapidly activated by IL-5, which allows Pin1 to interact with and isomerize GTPases, resulting in conformational change and increased GTPase activity towards downstream targets.

## Data Availability

The data presented in this study are available on request from the corresponding author.

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
