# Peer review of "Pin1 Regulates IL-5 Induced Eosinophil Polarization and Migration"

_cells, 2021, doi:10.3390/cells10020211_

Round 1
Reviewer 1 Report
Several major concerns:
- lack of validation of bone-marrow derived eosinophils for Pin1 expression: The Pin1 conditional knockout mice was creased by cross-breeding Pin1fl/fl with EoCre mice to generate mice with Pin1 knockout specifically in eosinophils. However, the eosinophils used in this study is bone-marrow-derived eosinophils (BMDEs), which may be different from the endogenous eosinophils. Thus, a confirmation is needed to see the expression of Pin1 in BMDEs from WT and EoCre; Pin1fl/fl mice.
- Less clear about the methods used for eosinophil polarization and migration. There are many known methods to detect cell polarization and migration. First, the rationale for the methods use is not clear, at least major references should be cited. Next, a bit more is needed to introduce the methods used, and lastly how these data were calculated and normalized.
- The interaction between Pin1 and RhoA was investigated. How about the interaction between Pin1 and Rac? What is the major functional axis, Pin1-RhoA or Pin1-Rac? or combination?
- It is critical to see the effect of Pin1 on eosinophil function (e.g., cytokine release). Thus, it would be of interest to see the differential cytokine/chemokine release from WT and Pin1-deficient eosinophils.
Reviewer 2 Report
The submitted paper by Shen et al is logical straightforward study that builds on the authors previous work. The authors use a number of in vitro experiments using human peripheral blood eosinophils or mouse bone-marrow derived eosinophils to demonstrate that IL-5 induced actin polarisation and migration is dependent on prolyl isomerase (Pin1), which acts upstream of the Rho family GTPases, especially RhoA. Only limited evidence is provided that Pin1 can act on other GTPases, RAC and CDC42.
My enthusiasm for the paper is tempered by its very low n# (n=3) for all experiments. A minimum of n = 5 should be considered in all experiments.
Please also provide information to whether technical replicates also performed.
I only have a general comments:
Please include the cell viability results in Fig 3
In Fig 4, the Rac inhibitor (RacIN) does not inhibit polarisation or migration in a dose-dependent manner. Furthermore, only one concentration for each inhibitor is shown in Fig 4B. Please rephrase and additionally replace the word dose with concentration.
Please provide methodological details for anti-pSer3-cofilin and anti-cofilin quantification by flow cytometry.
Please provide the quantification and high-resolution zoomed-in images for Pin1 and RhoA co-localisation in Fig 5B.
Why was no Rac (or even Cdc42) activity assay included in Fig 6A?
Minor
Please include full details on the used antibodies
Please provide company information for rmSCF and rmFLT3L
Please describe the DNA dye used in cell migration experiments and wavelengths used to detect it.
Please describe the post-hoc test used for statistical analysis.
Round 2
Reviewer 1 Report
My concerns have been well-addressed.